# Insights into Ultrasonication Treatment on the Characteristics of Cereal Proteins: Functionality, Conformational and Physicochemical Characteristics

**DOI:** 10.3390/foods12050971

**Published:** 2023-02-24

**Authors:** Yang Wang, Jiarui Liu, Zhaoli Zhang, Xiangren Meng, Tingxuan Yang, Wangbin Shi, Ronghai He, Haile Ma

**Affiliations:** 1College of Tourism and Cooking & College of Food Science and Engineering, Yangzhou University, Yangzhou 225127, China; 2School of Food and Biological Engineering, Jiangsu University, Zhenjiang 212013, China

**Keywords:** ultrasonication, cereal proteins, functionality, conformational, physicochemical characteristics

## Abstract

Background: It would be impossible to imagine a country where cereals and their byproducts were not at the peak of foodstuff systems as a source of food, fertilizer, or for fiber and fuel production. Moreover, the production of cereal proteins (CPs) has recently attracted the scientific community’s interest due to the increasing demands for physical wellbeing and animal health. However, the nutritional and technological enhancements of CPs are needed to ameliorate their functional and structural properties. Ultrasonic technology is an emerging nonthermal method to change the functionality and conformational characteristics of CPs. Scope and approach: This article briefly discusses the effects of ultrasonication on the characteristics of CPs. The effects of ultrasonication on the solubility, emulsibility, foamability, surface-hydrophobicity, particle-size, conformational-structure, microstructural, enzymatic-hydrolysis, and digestive properties are summarized. Conclusions: The results demonstrate that ultrasonication could be used to enhance the characteristics of CPs. Proper ultrasonic treatment could improve functionalities such as solubility, emulsibility, and foamability, and is a good method for altering protein structures (including surface hydrophobicity, sulfhydryl and disulfide bonds, particle size, secondary and tertiary structures, and microstructure). In addition, ultrasonic treatment could effectively promote the enzymolytic efficiency of CPs. Furthermore, the in vitro digestibility was enhanced after suitable sonication treatment. Therefore, ultrasonication technology is a useful method to modify cereal protein functionality and structure for the food industry.

## 1. Introduction

The global population is increasing and is forecasted to reach 10 billion by 2050. To meet the growing population, cereals, one of the most basic foodstuffs around the world, provide energy and nutrients to the global population [1]. Wheat, rice, barley, rye, oat, maize (corn), millet, and sorghum are the most common staple cereals [2]. The fact that cereals is a larger share of sustainable agriculture adds to the importance of cereal and cereal-based products. Cereal provides considerable carbohydrates, protein, dietary fiber, and bioactive nutrients that our bodies require for growth and metabolism [3]. Cereal proteins (CPs), a prior micronutrient of cereal, are gaining more attention from scholars working in the food industry because of their nutritional benefits and functional components.

CPs are primarily applied in food processing owing to their nutritious values and functional characteristics [4]. They are available for solubility, foaming, emulsifying, and gelling applications [5]. CPs, particularly gluten, are utilized in many products and byproducts, such as leavened and unleavened bread, noodles, and cookies [6]. However, the characteristics of proteins must be stable during mixing, heating, desiccating, storage, and physical, chemical, or enzymatic modifications, maintaining their structure and not damaging their physicochemical properties [7].

With the increasing demands of consumers for nutritious foodstuffs, emerging pretreatment technologies have been applied to food science such as ultrasonication, high-pressure processes, pulsed electric fields, microwaves, irradiation, low-temperature plasma, and radio frequencies. Studies such as these have drawn increasing attention [8]. These technologies play a critical role in maintaining food quality and safety with high nutritional merits and minimal damage over conventional food-processing techniques [9]. Ultrasonic technology has attracted much attention in the food industry, mainly because it is considered an innovative technology to achieve higher-quality and -functionality processed foods with ecofriendly chemistry and cost-effective industrialization [10]. Ultrasound is a mechanical wave at a higher frequency than that of the level of human hearing (>16 kHz) that can be categorized into two types, low-intensity high-frequency, and high-intensity low-frequency ultrasound. Low-intensity high-frequency ultrasound (100 kHz–1 MHz, power < 1 W/cm^2^) is most often used for nondestructive analytical food testing, quality evaluation, and process control to ensure the quality and safety of food. High-intensity low-frequency ultrasound (18 kHz–100 MHz, power > 1 W/cm^2^), on the other hand, is mainly used to physically or chemically alter the properties of food in food processing such as extraction, protein enzymolysis, peptides, and modifications [11,12,13]. Therefore, the ultrasonic technique is commonly utilized in food processing as it is a proven effective approach to modify protein properties via cavitation [12].

Studies were conducted on the influence of sonication on the physicochemical characteristics and structural features of CPs [14]. However, there has been no a systematic review concerning this subject. Therefore, the objective of this review is to winnow all the eligible articles from different databases via appropriate keywords and a comprehensive search. On the basis of the analytical and synthesis results, we summarize how sonication treatment affected the functional and structural characteristics of CPs. The review provides indepth information on sonication’s influence in CPs research and guide the food industry in making strategic decisions.

## 2. Search Program and Selection Standards

In this review, all related articles were retrieved through a comprehension search. There are several main hotspots related to ultrasound, ultrasonic practices, or ultrasonication in the databases with the subject classification of “food science technology”, such as “protein”, “high-intensity ultrasound”, “functional properties”, “structural characteristics”, and “physical properties”. On the basis of these articles, the co-occurrence network of numerous keywords in bibliographic data is exhibited from 2014 to 2022 via VOSviewer software (Figure 1). As shown in Figure 1, ultrasound has been widely used in the modification of CPs in recent years.

All related studies were selected after examining their title, abstract, and full content, and ultimately on the basis of inclusion and exclusion criteria. Inclusion criteria were as follows: (1) they reported on the functional, structural, and physicochemical characteristics, including solubility, foaming and emulsification, surface-hydrophobicity, sulfhydryl group, particle-size, conformational-structure, microstructural, enzymolytic, and in vitro digestive properties; (2) ultrasonic treatment was involved; (3) published date from 2000 to 2022; (4) no language limitations; (5) they included CPs. Moreover, the following were the exclusion criteria: (1) ultrasonic treatment not mentioned; (2) CPs were omitted; (3) publishing date before 2000. There was evidence that different ultrasonic conditions, such as power density, frequency, temperature, amplitude, and sonication time, could lead to different results. Consequently, the final results were systematically analyzed and summarized.

## 3. Results and Discussions

### 3.1. Changes in the Functionality of CPs via Ultrasound

#### 3.1.1. Solubility

The solubility of CPs, as one kind of functionality, plays a major role in the quality of food products. The solubility of proteins refers to the extent to which proteins were denaturized and aggregated. This may impact other functional characteristics, such as foaming and emulsifying properties [15]. Therefore, the other technofunctional characteristics of proteins and their application may be influenced by their solubility within the food industry. Reasonable ultrasonic conditions are used to enhance solubility up to a certain point. The influences of ultrasound on CP solubility can be ascribed to conformational variations in the (partial unfolding of the) protein structure caused by the implosion of cavitation bubbles. Physical perturbations help in breaking up hydrogen and hydrophobic bonds, exposing the buried hydrophilic groups to the surrounding water. Thus, the interaction between protein molecules and water is promoted, enhancing protein solubility [16,17].

A study by Wang et al. [18] demonstrated that, in comparison with the rice bran protein without ultrasonic treatment, the solubility of rice bran protein with ultrasound (20 kHz; 6 mm diameter titanium probe with a depth of 1 cm; URBP) had obviously been increased via the enhancement of ultrasonic power. The solubility of URBP reached the maximum when the ultrasonic power was 200 W and the ultrasonic time was 10 min.

Other findings regarding ultrasonic treatment of CPs exhibited a homotrend of an increase in protein solubility, such as protein isolates (BPIs) [19], rice dreg protein isolates (RDPIs) [20], sorghum kafirin [21], and oat protein isolates (OPIs) [22], via various types of ultrasonic devices. The decrease in the particle size and partial unfolding of proteins during ultrasonication increased the charged groups on the surface of the protein. This phenomenon also has significant influence on protein–water and protein–protein interactions via hydrogen bonds, hydrophobicity, and electrostatic forces, and as a result, causes protein dispersion and increases protein solubility. Figure 2 presents an illustration of the cavitation effect. It is also essential for maintaining a consistent temperature with ultrasonication when investigating protein dissolution [23]. Similar results utilizing sonication are reported in Table 1.

Nevertheless, studies have demonstrated that the solubility of CPs significantly decreases when they are exposed to excessive high-intensity ultrasound. For example, Chen et al. [30] observed the impact of the solubility of rice protein (RP)–dextran conjugates treated with ultrasound-assisted glycation (URPDCs) (25 kHz; power: 400, 500, and 600 W). When the ultrasonic power increased (under 700 W), the solubility increased significantly. The solubility reached up to 88.5% at 600 W and then reduced at ultrasonic power of 700 W. Cavitation generated by ultrasonic treatment likely enhanced the proteins’ solubility, whereas excessively high ultrasonic power could cause protein aggregation, restraining the reaction. Proteins were more susceptible to aqueous solutions because of the broken covalent bonds and more hydrophobic groups caused by acoustic cavitation. Therefore, larger aggregates are formed through hydrophobic bonds, destabilizing dissociated proteins and thus reducing solubility [40,41]. The majority of the available literature on ultrasonication regards the frequency range of 20–50 kHz and moderate power under 700 W; ultrasonic treatment may effectively promote the solubility of CPs.

#### 3.1.2. Emulsifying Properties

Emulsifying properties play a crucial role in the application of proteins as surfactant substances. Because of the amphiphilic nature of proteins, the solid homogeneous emulsion of a protein can be established in an oil–water system via surface active agents [42], thus improving the emulsification attributes, including the emulsifying activity index (EAI) and emulsion stability index (ESI). In addition, the modification of CPs’ emulsification properties dramatically influences the protein size, conformation, surface hydrophobicity, and molecule flexibility under an ultrasonic field [24,29]. Modifications in the emulsifying characteristics of an isolated cereal protein caused by sonication were evaluated. Cavitation efficiency may affect emulsification during sonication. Zhang et al. [43] significantly increased the EAI and ESI of wheat gliadin (WG) and green wheat gliadin (GG) treated with sonication with various power inputs. The increase in the emulsification characteristics of WG and GG via sonication was related to the smaller particles with the help of sonication acoustic cavitation. With the extension of acoustic time or/and ultrasonic power, the dispersed phase volume and bubble population increased, increasing the shear forces transferred through the rapid collapse of bubbles. Further, the disruption of oil droplets became more favorable, thereby strengthening emulsion stability [12,44].

In another study, Hu et al. [45] identified an association between protein secondary and tertiary structural variation, and improved the emulsifying properties. When α-helix and β-sheets are influenced by ultrasonication, the ultrasound-treated protein has a better effective potential for adsorption capacity on the interface of oil/water. The particle size was reduced with ultrasonication, which increased the ratio of the surface area to the volume, thus increasing the emulsification characteristics [46]. Further, the proteins’ surface hydrophobicity that is increased with ultrasonic treatment can also act as a driving force for a reduction in the tension on the oil–water interface. Hence, the protein absorption rate was increased, which helped in rendering the films rigid through hydrophobic/hydrophilic groups. As a result of these alterations, proteins treated with sonication may be more easily emulsified. Furthermore, high-intensity ultrasound can enhance emulsion homogenization by pretreating the proteins; it is widely applied in food processing. We show previous results from other authors in Table 1. The results show that ultrasonic treatment such as at a 20 or 25 kHz frequency and with power of 100 or 300 W could lead to a significant increase in emulsification efficiency.

In contrast, excessive ultrasonic power input might cause a loss in the emulsification properties of the protein. Wang et al. [29] found that the emulsification characteristics of rice bran protein decreased slightly as ultrasonic power increased between 450 and 600 W. The opposite might be attributed to the intensified sonochemical effects induced by excessive power disrupting the protein’s secondary structure, resulting in the flocculation of interfacial proteins and the aggregation of emulsion droplets. Therefore, it is important to select the appropriate ultrasonic power and frequency levels for different CPs to maintain an equilibrium between the exposure of hydrophobic/hydrophilic groups and protein aggregation to achieve excellent emulsification performance.

#### 3.1.3. Foaming Properties

Foaming characteristics are largely determined via molecular movement, penetration, and rearrangement at the air/water interface, mainly applied in food processing. Foaming capacity (FC) depends on the protein dispersion, unfolding, and repositioning at the gas/solvent interface to decrease tension at the interface, while foaming stability (FS) is mostly dependent on the formation of a cohesive, robust layer around air bubbles. There are verified relationships among the foaming properties, structural flexibility, and surface hydrophobicity of proteins [44]. After sonication, the rapid diffusion of molecules in the air/liquid interface and molecular rearrangement allow for cohesive viscoelastic films to entrap air, which can modify the foaming properties. On this basis, there are also close relationships between foaming properties and other properties, such as particle size, surface hydrophobicity, molecular weight, and structural flexibility [47].

Appropriate ultrasonic modes can improve the FC of CPs. Akharume et al. [34] reported that FC improved significantly under a long treatment for both the prolamin and glutelin fractions via sonication (20 kHz; 12.70 mm probe; 100%, 75%, 50% amplitude; 5, 10 min). FC was enhanced from 3.83 to 10.33% for the Dawn prolamin and from 22.50 to 34.33% for the plateau glutelin. In addition, FS increased from 57.50 to 100% at 52.72 W for 5 min. Comparable observations were reported in earlier studies: rice bran protein [19], millet protein concentrate (MPC) [14], and foxtail millet concentrates [25].

Study outcomes are listed in Table 1. Ultrasonication treatment enhanced the foaming properties with a frequency of 20 kHz and power range of 100–600 W. The observed improvement in the foaming characteristics was attributed to the cavitation effect of sonochemical action. The surface hydrophobicity and molecular flexibility of protein molecules were improved. The particles of proteins were also distributed more evenly, and particle size was reduced. These changes might have led to a rapid enhancement of the adsorption ability on the gas/liquid interface and thus resulted in greater foaming capacity. Furthermore, ultrasonic treatment could induce changes in the conformation of protein molecules, namely, the partial exposure of the protein structure and hydrophobic amino acid residuals to polymerize viscoelastic films at the air/water interface [48].

However, excessive ultrasonic treatment should be noted considering that the reaggregation of proteins induces the desorption of protein molecules at the air–water interface [24]. Meanwhile, the foaming properties of protein might be influenced by other sonication parameters, such as intensity (W/cm^2^) and power density (W/mL).

### 3.2. Surface Hydrophobicity (H_0_)

Hydrophobic interactions play an essential role because they are highly associated with the content of hydrophobic amino acids in the food system [37]. Hydrophobicity (*H_0_*) is partly responsible for the conformational structures of proteins and protein correlations, such as polar–nonpolar group interactions and complex formation [45,49]. The *H_0_* of the protein significantly increased as hydrophobic groups were exposed to ultrasonic cavitation, which impacted the functionality of the proteins. With the prolongation of ultrasonic treatment, the spatial structure of the protein changed, exposing more hydrophobic amino acid residues and increasing the surface hydrophobicity because ultrasonic waves can break the hydrogen bonds, electrostatic interactions, and hydration between protein molecules, exposing the hydrophobic groups buried inside the protein molecules.

For example, Zhou et al. [50] evaluated the functional impact of ultrasonication (25 ± 1 °C initial temperature; 30 min; power 600 W) on the surface hydrophobicity of defatted wheat germ protein (DWGP) and indicated that the fluorescence peak intensity (420–540 nm) of DWPG increased gradually from 63.7 to 573.25 W/cm^2^ via sonication. Wang et al. [18] also examined a similar finding and reported an increase in *H_0_* with ultrasound-treated rice bran protein. Later, Yang et al. [51] found that the increase in *H_0_* could have been attributed to the cavitation action of ultrasound that had generated the turbulent shear force, microflow, and other effects. The initially buried hydrophobic regions in the interior of the molecule were effectively revealed to the hydrophilic surrounding medium through strong cavitation effects. Further, cavitation actions can destroy the protein molecules, shrinking the particles from large aggregates into smaller fragments, hence improving the hydrophobic surface of the protein [48]. Thus, as demonstrated in the studies shown in Table 1, ultrasonic treatment with a frequency range of 20–40 kHz and power range of 80–400 W could enhance the *H_0_* of CPs to varying degrees.

### 3.3. Sulfhydryl (SH) and Disulfide Bond (SS) Content

The sulfhydryl (SH) and disulfide (SS) groups are widely acknowledged as critical functional groups in protein molecules. Both play essential roles in maintaining the stability of protein structures, and their ratio can also influence the functionality of proteins [37]. The modification of the free sulfhydryl content of protein molecules could be directly related to the denaturation degree of proteins. For example, Yang et al. [52] found that, at ultrasonic power intensity from 40 to 100 W/L, the SH of defatted wheat germ protein increased significantly (*p* < 0.05). The highest increment in SH at 60 W/L was 53.20 μmol/g, an increase of 43.21%, which remained steady with increasing ultrasonic power density. The increase in SH can be attributed to the stretching of the protein and its internal sulfhydryl group being exposed to its external surface. The possible reason was that the buried sulfhydryl groups could be unfolded, accompanied by a reduction in protein size, disrupted by the high pressure and shear force caused by sonication [51]. Later, another study by Qin et al. [37] reported that the number of free SH groups in soy protein isolate/wheat gluten mixture increased during the dual modification of protein under high-intensity ultrasound and microbial transglutaminase (MTGase) cross-linking. Similarly, this trend is consistent with the work by Zhang et al. [43] for wheat gliadin. Their study indicated that SH content reached the maximum (10.99 μmol/g) with ultrasonic power of 400 W.

Another study, by Zhang et al. [20], examined the effect of sonication (20, 28, 35, 40, and 50 kHz; power density 400 W/L) on the SH and SS of rice dreg protein isolates. The results showed that sonication treatment caused an increase in the total sulfhydryl group and free sulfhydryl content, and a decrease in SS bonds, especially at a frequency of 20/40 kHz. Later, Liu et al. [35] studied the effect of sonication time (5, 10, 15 min) and power (130, 160, 200 W) on the SH and SS of yellow dent corn-separated protein. The works indicated that the SH content increased by 37.21%, and the SS content decreased by 43.66% via sonication (15 min, 200 W). This was attributed to ultrasonic cavitation, which induced the sulfhydryl group to be exposed to the protein surface. In other words, CPs treated with sonication may exhibit an increase in SH due to the oxidation of hydroxyl radicals generated by cavitation. Meanwhile, the disintegration of SS destroyed the protein conformation and hydrophobic groups initially inside, and the protein molecule was exposed to the external surface more [53].

### 3.4. Particle Size

The particle size of proteins is an essential factor influencing protein functionality. A (controlled or moderate) sonication treatment could reduce the particle size of CPs through protein aggregations. The smaller particles of proteins could be attributed to the disruptive effects of sonication acoustic cavitation. Cavitation damages the electrostatic interactions and hydrogen bonds between CPs molecules, resulting in protein molecules aggregating into smaller fragments [29,54].

Similar study outcomes of CPs via sonication displayed a decrease in the particle size under various ultrasonic conditions. For example, as reported by Sharma et al. [25], ultrasonic treatment (amplitude: 5 and 10%; duration: 5, 10, and 20 min) significantly decreased (*p* < 0.05) the particle size of foxtail millet protein, and the decrease continued with the increase in ultrasonic time. Moreover, Zhang et al. [43] indicated that, as the ultrasonic power was from 0 to 400 W, the average particle size of wheat gliadin and green wheat gliadin by sonication was reduced by 42.1% and 32.2%, respectively. Further, O’Sullivan et al. (2016) [31] also found that ultrasonic treatment (20 kHz, 34 W/cm^−2^ for 2 min) could significantly reduce the particle size of wheat protein. The tendency was in concordance with the reports by Qin et al. [37], Sun et al. [19], Wang et al. [18], and Jin et al. [38]. In addition, when ultrasonic treatment is combined with other treatments, the particle size of proteins might be altered. For instance, Zhang et al. [20] concluded that the particle size of rice dreg protein isolates decreased from 330.8 to 219.6 nm after ultrasound-assisted alkali treatment.

However, various trends in the particle size of CPs were reported in several studies. Wang et al. [29] found that, as ultrasonic power increased to 500 W, there was an increase in the particle size of rice bran protein. The phenomenon may have been due to the reaggregation of small particles through the thermal effect generated by the ultrasound. This result agreed with Jiang et al. [55], who confirmed an increase in the particle size of black bean protein isolates via ultrasonication (20 kHz, 450 W). The enhancement in particle size might be related to the repolymerization of aggregates through noncovalent and covalent interactions. In general, when ultrasound is applied, appropriate frequency and power parameters could reduce the particle size of CPs. The particle size might have been enhanced, but ultrasonic power was excessive [31].

### 3.5. Conformational Structures

In addition to modifying the functional characteristics properties of the protein, ultrasonication can alter its structural characteristics. In accordance with the progressive state of the spatial arrangement of polypeptide chains, the protein structure is categorized into primary, secondary, and tertiary structures. The protein structure is differently affected by ultrasonic treatment on the basis of the types and conditions of ultrasonic treatment. As shown in Figure 3, when the treated protein was exposed to ultrasonic cavitation, alterations only happened in the secondary and higher-order structure of the protein except for the primary [56]. Once the noncovalent interactions between proteins and polysaccharides are modified by cavitation and shearing forces, and hydrophobic and hydrogen bonds are broken, thereby contributing to a variety of corresponding conformational changes (such as unfolding, denaturation, and reaggregation) and sequential alterations in technofunctional and nutritional properties [19,57].

#### 3.5.1. Primary Structure

Proteins are composed of amino acid sequences as their primary structure. Sodium dodecyl sulphate–polyacrylamide gel electrophoresis is the main method used to examine the changes in subunits of CPs following sonication. According to Wang et al. [11], there was no noticeable alteration in the subunit of rice dreg protein isolates via sonication (20, 28, 35, 40, 52 kHz). This phenomenon was attributed to alterations in the molecular weight of the proteins, which was not reflected in the electrophoretic spectrum of the protein. Li et al. [22] found that sonication did not alter oat protein composition when the oat protein was subjected to high-intensity ultrasound (20 kHz, 80 W) for 5 min at 70% amplitude. However, some studies are in disagreement with these viewpoints. For example, Jhan et al. [59] showed that the main polypeptide band was around 20 kDa for all native proteins. and that the molecular weight (27–10 kDa) of sorghum protein nanoparticles was reduced following sonication-assisted nanoreduction. This might be attributed to the breakdown of intermolecular hydrogen and hydrophobic bonds, decreasing the molecular weight of the proteins. Furthermore, Nazari et al. [14] indicated that there was a reduction in the molecular weight (40–50 kDa) of millet protein concentrate after ultrasonic treatment (20 kHz, 73.95 W/cm^2^ for 12.5 min). So, sonication has various effects on the primary structure of proteins due to various ultrasonic conditions.

#### 3.5.2. Secondary Structure

The secondary structure of the protein is formed by primary polypeptides on a nascent protein in distinctively coiled aqueous environment. The secondary structure of proteins is characterized through the ratios of α-helix, β-sheet, β-turn, random coil, and unordered groups. Those movements could be transformed by each other to some extent after ultrasonic treatment. Therefore, sonication could alter the protein secondary structure. For instance, Sullivan et al. [21] reported that there was an increase in the amount of unordered or random coils of purified kafirins, while the α-helix content decreased after ultrasonic treatment (20 kHz ± 50 kHz at 40% amplitude for 10 min). In addition, the work of Zhang et al. [60] concluded that the α-helical content of gluten protein was decreased, while the β-sheet and β-turn contents were increased after multifrequency sonication treatment (28, 40, and 80 kHz).

A similar phenomenon was studied by Wang et al. [27] regarding corn gluten meal protein, and Liu et al. [35] regarding yellow dent corn separated protein. In the former study, low-power density ultrasound (20/40 kHz, 100 W/L, 20 min, 5:2 s/s) altered the secondary conformational structure of corn gluten meal protein, leading to a reduction in α-helix and β-turn, and an increase in β-sheet and random coil. In the latter study, there was a decrease in α-helix and β-turn, and an enhancement in β-sheet and random coil via sonication (200 W, 15 min), resulting in transforming the yellow dent corn separated protein structure from order into disorder. Furthermore, these observations are in line with earlier studies on wheat gluten by Qin et al. [37], and Zhang et al. [61].

On the basis of the aforementioned studies, even though the secondary structure might be influenced by the sonication device, intensity, time, and frequency, a protein could undergo some changes in the secondary structures, and exhibit a looser and more flexible structure following ultrasonication induced by cavitation.

#### 3.5.3. Tertiary Structure

The protein’s tertiary structure is preferred three-dimensional arrangement of the folded polypeptide chains. Compared with a protein secondary structure, a tertiary structure directly affects the functional characteristics of the proteins. Tryptophan residues are essential indicators for characterizing the tertiary structure [13,62]. Alterations in the tertiary structures of wheat gliadin (GG) and green wheat gliadin (WG) with different ultrasonic conditions were investigated by Zhang et al. [43]. By means of fluorescence spectroscopy, they found that the fluorescence intensity of GG and WG decreased significantly after ultrasound. This might be ascribed to the bubbles transferring through the sound wave, and the location was instantaneously heated. Cavitation generated by the bubble burst could have unfolded and exposed the protein structure during ultrasonic treatment. Moreover, a slight red shift was observed in the maximal fluorescence emission wavelengths of GG and WG: 3 and 2 nm, respectively. In this experiment, an increase in the polarity of the tryptophan residue microenvironment was demonstrated, thereby indicating that ultrasonic treatment enhanced the tertiary structure formation of the GG and WG molecules. A similar conclusion was reached by Su et al. [63], who found that mass transfer effects enhanced by ultrasound might cause transient bubbles. When the bubbles collapsed, vast reactive free radicals could have led to the modification of amino acid side chains, which would change three-dimensional folded structures. Therefore, ultrasound has the ability to irreversibly alter and adapt the three-dimensional structure of CPs to emulsify oils, yielding a stable structure for up to years. Qu et al. [64] also mentioned that ultrasound-treated (20/28 kHz, 150 W/L) CPs had shown increased absorption intensity at 275 nm. and a new structure had been shaped. Thus, ultrasonic working parameters such as ultrasonic power, time, and temperature might influence the tertiary structure of proteins.

### 3.6. Microstructure

Having undergone ultrasonic treatment, the structure of CPs was reduced from large and interconnected aggregates to small and dense fragments with loose and granular distribution [65]. The microstructure of CPs was influenced according to the cereal protein type and source, besides ultrasonic parameters and processing conditions. For instance, Jin et al. [38] reporte that the turbulent and cavitation forces could fracture the macroparticles and further change the surface state of the proteins. According to this study, ultrasonication caused cavitation bubbles and microstreaming effects that disrupted the formation of protein aggregates. The aggregation of fragments might be attributed to the destruction of cross-links between amino acid residues in protein containing S–S bonds, hydrophobic bonds, and Van der Waals interactions generated by ultrasonic radiation [11,59]. A similar phenomenon was observed by Zhang et al. [66]. The researchers studied the effects of alternate dual-frequency ultrasound on the structure of wheat gluten, and found that sonication caused alterations in the structure, which displayed irregular agglomerates. The findings from their work indicated that the values of roughness (*R_a_*, *R_q_*) greatly increased with sonication (20/35 kHz, 150 W/L, 10 min). Likewise, Wang et al. [11] demonstrated that sonicated rice dreg protein isolated with multifrequency countercurrent S-type sonication displayed more uniform protein fragments and a looser structure under ultrasonic conditions (20/40 kHz, 60 W/L, 20 min). Their findings also agreed with the results of rice proteins obtained by Yang et al. [51], and Wang et al. [27].

Apart from the modification of CPs via ultrasonic treatment alone, ultrasound combined with alkali treatment, one of the synergistic methods of ultrasonic modification aided by other chemical and biological means, has gained considerable attention. For example, Zhang et al. [20] observed that, after sonication-assisted alkali treatment, the structure of rice dreg protein isolates loosened and disordered, exhibiting more irregular fragments and microparticles. Li et al. performed a similar study [67]. The alterations in the microstructure may be ascribed to the generation of cavitation bubbles that disrupt protein aggregation, thereby destroying the cross-linking reaction between protein molecules. Meanwhile, variation in the morphological structures might be attributed to the unfolding and partial denaturation of the tertiary protein structure, liberating more hydrophobic bonds, thereby leading to different changes in the CPs’ functionality.

### 3.7. Enzymatic Hydrolysis

Enzymatic hydrolysis is broadly applied to enhance the functional quality of proteins, which is beneficial to biological substances in the human body. Compared to traditional enzymatic hydrolytic technology, ultrasound-assisted enzymolysis has better advantages, such as easier control, shorter processing time, and more convenient operation [68]. Ultrasonic pretreatment could generate special acoustic cavitation that produces high-intensity shearing force, free radicals, and shock waves, leading to protein denaturation to release hydrophilic groups, thereby affecting the solubility, bioactivity, and enzymatic efficacy of proteins [67,68].

A study by Qu et al. [69] examined the influence of sweep frequency and pulsed ultrasound (24 ± 2 kHz, 24 W) on the enzymatic hydrolytic efficiency and activity of ACE inhibitory peptides from wheat germ protein. In that study, ultrasound-assisted enzymatic hydrolysis could change a conformational molecule and improve the efficiency of enzymatic hydrolysis. This observation indicated that ultrasound could increase enzymatic activity on the basis of reasonable conformational alterations in the protein. These results agree with those of Li et al. [70], who indicated that sonication (28 kHz; 2 cm deep inserted with ultrasonic probe) could shorten the enzymolytic action time of rice protein (RP) and improve the efficiency of enzymatic hydrolysis. Compared with untreated RP, the structural surface of enzymolytic residues became uniform with many small fragments. Alterations in the structures of hydrolytic residues via sonication resulted in the improvement of enzymatic efficacy. The results agreed with those by Li et al. [39,71] for rice protein.

Similarly, a recent study by Wang et al. [27] found that the enzymolytic efficiency of corn gluten meal was enhanced via sonication (sequential double frequency of 20/40 kHz). Enzymolytic efficiency reached a maximum of 15.99% with a protein dissolution rate of 61.69%. The mechanism could be ascribed to the collapse of cavitation bubbles formed by sonication at a frequency of 20 kHz, providing new cavitation nuclei for the ultrasound at 40 kHz, ultimately resulting in an increase in cavitation bubbles. Figure 4 is a diagram of the whole process of ultrasonically treated corn gluten meal. Jin et al. [71] used ultrasonic sweeping frequency (28 ± 2/68 ± 2 kHz, 80 W/L, 40 min) to study the enzymolysis of corn gluten meal, and their results indicated that ultrasound could accelerate enzymolysis, leading to an increase in the affinity between enzyme and substrate. Further, a similar phenomenon was shown after ultrasonic treatment in other CPs, such as defatted corn germ protein [72], defatted wheat germ protein [52], and oat-isolated protein [73]. Therefore, whether ultrasonication conditions are a single, dual, or triple ultrasonic frequency, ultrasound can be considered a promising treatment technique for enhancing the enzymatic efficacy of CPs.

### 3.8. In Vitro Digestion

The digestibility of protein mainly reflects the extent to which protein in food has been absorbed and utilized by the human body in the digestive tract. Food protein digestibility is related to the susceptibility of a protein to proteolysis. Increased digestibility meant that it could be better hydrolyzed by pepsin and trypsin, and have higher nutritional value. According to previous studies, ultrasound-assisted treatment is commonly used to optimize CPs and CP-based products so as to increase their digestive value within the human gut.

For instance, Hassan et al. [74] investigated the influence of ultrasonically treated sorghum grain protein on in vitro digestibility. In that study, ultrasound (40% amplitude for 5 min) showed significantly higher digestibility (64.70 ± 0.50%) than that of other ultrasonic conditions (60% amplitude for 10 min). Compared with the control, germination was significantly promoted by sonication, thereby improving protein digestibility. Moreover, Li et al. [39] investigated the influences of various ultrasonication working modes on the in vitro digestibility of rice protein. According to that study, mono-, dual-, and triple-frequency ultrasonication significantly improved the simulated gastrointestinal digestion product. An increase in hydrolyzed protein content promoted the contact activity of an enzyme involved in proteolysis, subsequently boosting the simulated gastrointestinal digestion process.

In another study, wheat gliadin and green wheat gliadin treated with ultrasound (400 W, 300 W) improved in vitro digestibility by 12.75% and 11.03%, respectively [43]. Similarly, Jin et al. [38] showed that sonication (20 kHz, pulsed on-time 10 s/off-time 5 s, amplitude 60% for 10 min) enhanced the in vitro digestibility of buckwheat protein, which increased by 41% compared to the untreated samples. This trend could be attributed to the alternations in protein structure by acoustic cavitation effects. Nevertheless, high-intensity ultrasonic treatment could decrease the in vitro digestibility of proso millet bran protein when the ultrasonic conditions were a frequency of 24 kHz, power of 400 W, and amplitude of 100% for 20 min. The decrease in the in vitro digestibility was caused by the temperature increase during high-intensity ultrasonic treatment. Generally, moderate sonication could influence the digestibility and nutritional quality of CPs.

## 4. Conclusions and Future Trends

Sonication treatment has many advantages in terms of enhancing the functionality, conformation, enzymolysis, and digestive properties of CPs. As discussed in this review, the modification of CPs with ultrasonication technology mainly improved the functional, conformational, and physicochemical characteristics of CPs. In this regard, sonication could alter protein structural characteristics, including surface hydrophobicity, particle size, conformational (primary, secondary, and tertiary) structures, and microstructures, to improve the functionality of CPs. Moreover, alterations in the protein molecules with ultrasonic acoustic cavitation can influence the physicochemical properties, such as bu improving enzymatic efficacy and in vitro digestibility. In addition, the impact of sonication on the CP structure are related to the type of ultrasonic device/equipment, ultrasonic treatment conditions, and processing intensity. Therefore, the specific mechanism of ultrasonic processing technology or various types of ultrasonic equipment on CPs should be further studied. At the same time, it is necessary to recognize the dynamic change process of CP structures via sonication through new technologies, and precisely control the spatial structure, which is conducive to improving the functionality of CPs. Lastly, research on large-scale industrialization should also be strengthened.

## Figures and Tables

**Figure 1 foods-12-00971-f001:**
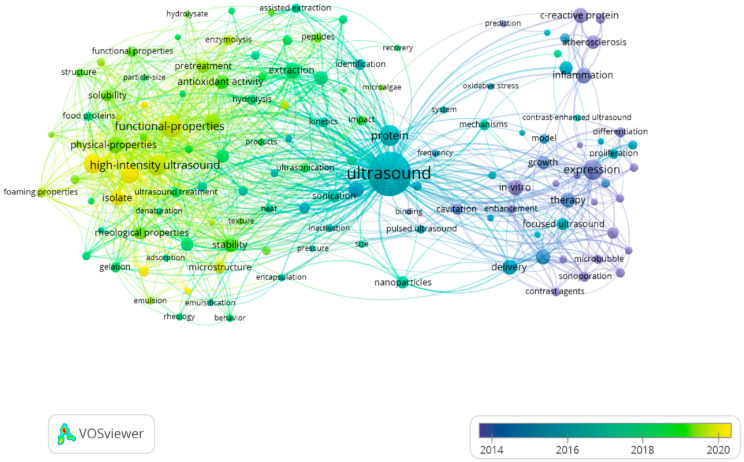
Visualization of trends in the modification of cereal proteins with ultrasound for enhancing functional properties.

**Figure 2 foods-12-00971-f002:**
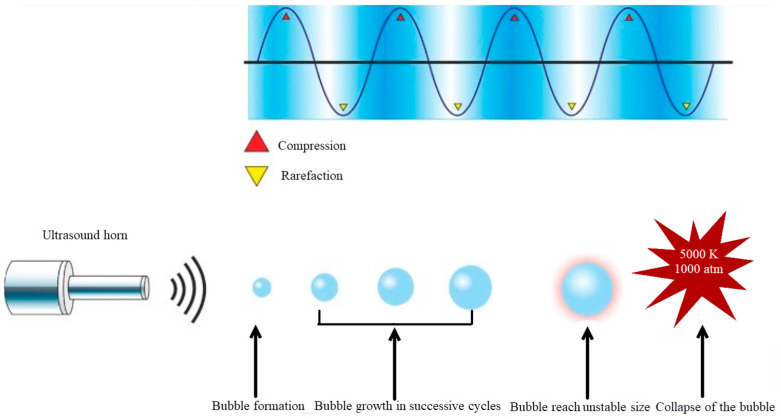
Schematic illustration of the ultrasonic generation of bubble cavitation and collapse [24].

**Figure 3 foods-12-00971-f003:**
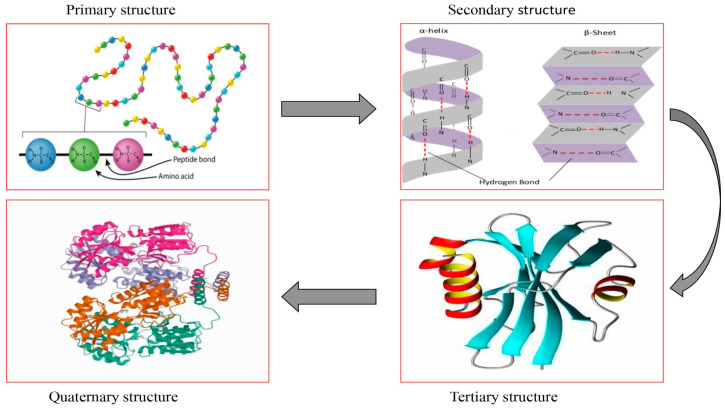
Sequential alterations in the primary, secondary, tertiary and quaternary structures of a protein exposed to ultrasonic treatment [58].

**Figure 4 foods-12-00971-f004:**
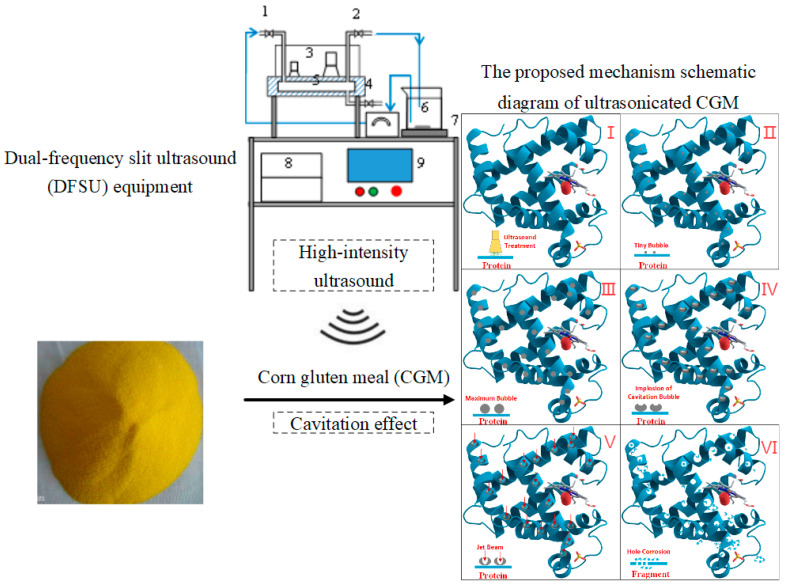
Diagram of the whole process of ultrasonically treated corn gluten meal (CGM) [41]. Note: The proposed mechanism schematic diagram of ultrasonicated CGM: (I)-Potein by ultrsound; (II)-Formation of tiny bubbles; (III)-Formation of cavitation bubbles; (IV)-Implosion of cavitation bubbles; (V)-Generation of jet beam; (VI)-Hole corrosion. Dual-frequency slit ultrasound (DFSU) equipment: 1. Sample inlet; 2. Sample outlet; 3. Ultrasonic transducer; 4. Waste outlet; 5. Slit cavity; 6. Treat liquid; 7. Magnetic stirrer; 8. Ultrasonic generator; 9. Programmable Logic Controller (PLC) control panel.

**Table 1 foods-12-00971-t001:** Summary of representative research on the functionality changes in modified CPs after ultrasonic treatment.

Proteins	Ultrasonic Conditions	Solubility	Emulsibility	Foamability	Surface Hydrophobicity	References
Foxtail millet protein concentrate	A 1 cm diameter probe; amplitude: 5, 10%; duration: 5, 10, 20 min; room temperature.	Increased	Increased	Increased	Increased	Sharma et al. (2022) [25]
Barely protein isolate (BPI)	A 13 mm diameter stainless steel probe; frequency: 20 kHz; amplitude: 100%;	Increased	—	—	—	Silventoinen et al. (2020) [26]
Corn gluten meal protein (CGM)	frequency: 20/40 kHz; power density: 60, 80, 100, 120, 160, 200 W/L; time: 10–40 min; temperature: 20–45 °C; pulse period: 5:1–5:5 s/s	Increased	—	—	Increased	Wang et al. (2020) [27]
Rice protein (RP)	Frequency: 25 kHz; power: 400, 500, 600 W; time: 15, 20, 25 min; temperature: 75, 80, 85 °C.	Increased	—	—	—	Chen et al. (2022) [28]
Rice bran protein (BRP)	A 6 mm diameter titanium probe; frequency: 20 kHz; time: 10, 20 min; power:100, 200 W.	Increased	Increased	—	—	Wang et al. (2021) [18]
Millet protein concentrate (MPC)	A 3 mm diameter titanium sonotrode probe; frequency: 20 kHz; power: 100 W; amplitude: 20%, 60%, 100%; time: 5, 12.5, 20 min; temperature: 20–30 °C.	Increased	Increased	Increased	—	Nazari et al. (2017) [14]
Oat protein isolate (OPI)	Frequency: 20 kHz; power: 80 W; amplitude: 70%; time: 5 min; temperature: 20 °C	Increased	—	—	Increased	Li et al. (2021) [22]
Sorghum kafirin	Frequency: 20 kHz ± 50 kHz; time: 5, 10 min; temperature: 4 °C.	Increased	—	—	—	Sullivan et al. (2018) [21]
Rice bran protein hydrolysate (RBPH)	Frequency: 20 kHz; pulse period: 6 s (4 s on, 2 s off); power: 0, 150, 300, 450, 600 W.	—	Increased	—	—	Wang et al. (2022) [29]
Rice protein (RP)	Frequency: 25 kHz; power: 400, 500, 600 W; time: 15, 20, 25 min; temperature: 75, 80, 85 °C.	—	Increased	—	—	Chen et al. (2022) [30]
Wheat protein isolate (WHPI)	Frequency: 20 kHz; amplitude: 95%.	—	Increased	—	—	O’Sullivan et al. (2016) [31]
Quinoa protein isolate (QPI)	Power: 100 W; temperature: 25 °C; time: 20 min	—	Increased	—	—	Xin-Sheng Qin et al. (2018) [32]
Rice bran protein	Power: 100, 200, 300 W; time: 10, 20 min	—	—	Increased	—	Sun et al. (2021) [19]
Rice protein (RP)	Frequency: 20 kHz; power: 600 W; temperature: 50 °C; time: 60 min	—	—	Increased	—	Zhang et al.(2017) [33]
Proso millet protein	Amplitude: 50%, 75%, 100%; frequency: 20 kHz; time: 5, 10 min.	Increased	—	—	—	Akharume et al. (2022) [34]
Yellow dent corn separated protein	Power: 130, 160, 200 W; time: 15 min	—	—	—	Increased	Liu et al. (2021) [35]
Rice bran protein (RBP)	Frequency: 20 kHz; pulse period 6 s (4 s on, 2 s off)	—	—	Increased	—	Wang et al. (2021) [36]
Rice dreg protein isolates (RDPI)	Frequency: 20, 28, 35, 40, 50 kHz; power density: 400 W/L	—	—	—	Increased	Zhang et al. (2020) [20]
Wheat gluten (WG)	Frequency: 40 kHz; power: 400 W; time: 0, 10, 20, 30, 40 min	—	—	—	Increased	Qin et al. (2016) [37]
Buckwheat protein isolates (BPIs)	Frequency: 20 kHz; amplitude: 60%; duration time: 10 min	—	—	—	Increased	Jin et al. (2020) [38]
Rice protein	Frequency: 28 kHz; power density: 58 W/L; temperature: 50 °C	—	—	—	Increased	Li et al. (2016) [39]

Note: —, the corresponding functionality and surface hydrophobicity was not mentioned in the paper.

## Data Availability

No new data were created or analyzed in this study. Data sharing is not applicable to this article.

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
