# Peer review of "Insights into Ultrasonication Treatment on the Characteristics of Cereal Proteins: Functionality, Conformational and Physicochemical Characteristics"

_foods, 2023, doi:10.3390/foods12050971_

Round 1

Reviewer 1 Report (Previous Reviewer 1)

The manuscript has been improved and I appreciate the authors' efforts to respond my comments. The article has been reinforced mentioning the conditions of the studies consulted.

Reviewer 2 Report (Previous Reviewer 2)

The authors have made the indicated modifications of all the reviewers and the article has improved substantially. 

This manuscript is a resubmission of an earlier submission. The following is a list of the peer review reports and author responses from that submission.

Round 1

Reviewer 1 Report

Manuscript ID: foods-2069336

Title: Insights into Ultrasonic Treatment on the Properties of Cereal Proteins: Functional, Structural and Physicochemical Properties

Comments:

The manuscript is very interesting and presents novel, concise and well-structured information on the effect of ultrasound on the properties of cereal proteins.

In my opinion, the authors should clearly specify the effect of ultrasonic parameters or conditions on each of the properties studied. For example, regarding the effect that ultrasound has on the solubility of cereal proteins, the authors clearly indicate the studies consulted (Table 1), the study conditions, and the reported findings. They very briefly indicate the effect of ultrasonic power based on a single study (Chen et al., 2022, line 125) but do not compare with other consulted studies (Table 1).

They do not include a brief conclusion at the end of each section regarding the effect of the ultrasonic parameters (for example the ultrasonic system, some range of different power, intensity, frequency, time or temperature) that have effect on the studied property (in this case, solubility).

If the authors compare the studies consulted and indicate the effect of each ultrasonic parameter at each experimental condition studied, future studies could be more easily oriented to implement ultrasound technology at an industrial level in the short term. I suggest doing the same with the other sections (3.3 to 3.10).

Author Response

Response to reviewer #1's comments

The manuscript is very interesting and presents novel, concise and well-structured information on the effect of ultrasound on the properties of cereal proteins.

Questions: In my opinion, the authors should clearly specify the effect of ultrasonic parameters or conditions on each of the properties studied. For example, regarding the effect that ultrasound has on the solubility of cereal proteins, the authors clearly indicate the studies consulted (Table 1), the study conditions, and the reported findings. They very briefly indicate the effect of ultrasonic power based on a single study (Chen et al., 2022, line 125) but do not compare with other consulted studies (Table 1).

They do not include a brief conclusion at the end of each section regarding the effect of the ultrasonic parameters (for example the ultrasonic system, some range of different power, intensity, frequency, time or temperature) that have effect on the studied property (in this case, solubility).

If the authors compare the studies consulted and indicate the effect of each ultrasonic parameter at each experimental condition studied, future studies could be more easily oriented to implement ultrasound technology at an industrial level in the short term. I suggest doing the same with the other sections (3.3 to 3.10).

Answer: Thank you for your constructive criticisms. According to the suggestion, we added a brief conclusion at the end of each section regarding the effect of the ultrasonic parameters that have effect on the studied property. The contents were showed by red font. (Section 3.1 to Section 3.10)

Reviewer 2 Report

This paper (review) describes the insights into ultrasonic treatment on the properties of cereal proteins: functional, structural and physicochemical properties. The article is quite complete, it is of interest to the scientific community, it addresses the study of the effect of ultrasound in various aspects, the discussion is clear and sufficient, and gives a global vision of the application of focused ultrasound to cereal proteins.. The work is well discussed and is supported by the references provided by the authors. The English language is correct. The authors have a great knowledge of the subject as it is observed in the bibliography, and this work deepens even more in this field. This review is interesting and delves in the different uses of ultrasounds respect cereal proteins.

Lines 65, 66, 76, 87,…..: Capitalize each word according the format of the journal. Unify and apply to the entire document.

Lines 70, 121, 279, 391, ….: Use “Figure” instead of “Fig.” according the format of the journal. Unify and apply to the entire document.

Table 1, table 2, …Lines 240, ….: Unify “p < 0.05”. Put “p” in italics. Put a separation after and before “<” or “≤”. Unify and apply to the entire document.

Table 1, table 2, ….: Put a separation between a number and “ºC”. Unify and apply to the entire document.

Table 1, table 2, ….lines 195, ,….: Put a separation between the number and the units (W; kHz; Hz, min, etc). Unify and apply to the entire document.

Table 1, table 2, ….line 220, , ……: Put a separation after and before “=”, “<”, “±”. Unify and apply to the entire document.

Table 3: Use “mL” instead of “ml”. Unify and apply to the entire document.

Table 3, table 4, ….: Do not put a separation between a number and “%”. Unify and apply to the entire document.

Line 333: Check “[77…]”.

References: The name of the journals must appear abbreviated according to the format of the journal.

Reference 6 and 33: Check these references. Information is missing.

Author Response

Response to reviewer #2's comments

Q1: Lines 65, 66, 76, 87,…..: Capitalize each word according the format of the journal. Unify and apply to the entire document.

Answer: Thanks for pointing out the mistakes and kind advice. We have reviesed capital of each word according to the format of the journal in our full text, which were showed by green font. (Lines 92, 93, 102, 113, 125, 176, 177, 227, 265, 295, 296, 322, 344, 413,449, and 489.)

Q2: Lines 70, 121, 279, 391, ….: Use “Figure” instead of “Fig.” according the format of the journal. Unify and apply to the entire document.

Answer: Thanks for your review and kind advice and pointing out the mistakes. We have used “Figure” instead of “Fig.” in our full text, which were showed by green font.

Q3: Table 1, table 2, …Lines 240, ….: Unify “p < 0.05”. Put “p” in italics. Put a separation after and before “<” or “≤”. Unify and apply to the entire document.

Answer: Thank you for your carefully checking on our manuscript. We have modified “p” in italics and put a separation after and before “<” or “≤”. Italic “p” s were showed by green font. (Table 1, Table 2, Table 3, Table 4, Table 5, Table 6, Table 8, Table 10, Table 12. Lines 303, 332, 333,499,501.)

Q4: Table 1, table 2, ….: Put a separation between a number and “ºC”. Unify and apply to the entire document.

Answer: Thanks for your review and kind advice. We have put a separation between a number and “ºC” (Table 1, Table 2, Table 3, Table 4, Table 5, Table 6, Table 7, Table 8, Table 9, Table 10, Table 11, Table 12.)

Q5: Table 1, table 2, ….lines 195, ,….: Put a separation between the number and the units (W; kHz; Hz, min, etc). Unify and apply to the entire document.

Answer: Thank you for your carefully checking on our manuscript. We have revised the format in the Table and in the manuscript. (Table 2, Table 3, Table 4, Table 5, Table 6, Table 7, Table 8, Table 9, Table 10, Table 11, Table 12.)

Q6: Table 1, table 2, ….line 220, ……: Put a separation after and before “=”, “<”, “±”. Unify and apply to the entire document.

Answer: Thank you for your carefully checking on our manuscript. We have revised the format in the Table and in the manuscript. (Table 1, Table 2, Table 3, Table 4, Table 5, Table 6, Table 7, Table 8, Table 9, Table 10, Table 11, Table 12.)

Q7: Table 3: Use “mL” instead of “ml”. Unify and apply to the entire document.

Answer: Thanks for your review and kind advice and pointing out the mistakes. We have used “mL” instead of “ml” in Table 3, which were showed by green font.

Q8: Table 3, table 4, ….: Do not put a separation between a number and “%”. Unify and apply to the entire document.

Answer: Thanks for your review and kind advice and pointing out the mistakes. We have not put a separation between a number and “%” in our full text.

Q9: Line 333: Check “[77…]”. References: The name of the journals must appear abbreviated according to the format of the journal.

Answer: Thanks for your review and kind advice. We have checked the References 77, and the name of the journals have revised to abbreviated form according to the format of the journal. (References)

Q10: Reference 6 and 33: Check these references. Information is missing.

Answer: Thanks for your review and kind advice. We have checked all the references, and found that Reference 8 and 33 is missing information. We have added the information about Reference 8.

However, the reference 33 is incomplete. More literatures see oi:10.1007/s11694-022-01619-4. All references have revised and listed by green font.

Reviewer 3 Report

Totally speaking, this study is well designed. Firstly, this paper briefly discussed the ultrasonication effects on the enhancement of physicochemical, microstructure, and techno-functional characteristics of the protein, particularly focusing on the cereal proteins such as corn, barley, and sorghum. Elsevier Science, Web of Science, Wiley, and Spring-Linker databases were referred to search all connected articles from 2000 to 2022. The presented and discussed results have indicated that ultrasonic parameters and the type of sonication could affect (increase or decrease) the functional, structural, enzymatic, and digestive properties of cereal proteins. This study was suitable for this journal. However, there are two issues needed to be clarified.

(1) Lines 6 and 7: Please include the matching e-mail addresses of the co-authors listed above, as well as their initials in parenthesis.

(2) Specify the results acquired from this study of literature data in the work's Abstract. As a nutshell, the entire abstract is theoretical. For example, to identify a set of variables that lead to enhanced functional properties and conformational changes in one kind of protein molecule. Also, which type of sonication is best for causing beneficial modifications in cereal proteins?

(3) Line 53: I suggest that the authors use the term "high-intensity ultrasound" or "sonication technology" because it has been proven that only high-intensity, low-frequency ultrasound can lead to structural changes in proteins at the quaternary and tertiary structure levels.

(4) High-intensity ultrasound must be explained in more detail in the Introduction section, along with what kind of technology it is (frequency range, power range, and amplitude range) and the devices that are utilized to achieve it. It would also be beneficial to discuss some of the drawbacks of other non-thermal technologies (such as microwaves or high pressure) as well as the benefits of ultrasonic technology.

(5) Lines 90-91: Why did the authors decide to use only the 181-paper Web of Science indexed citation database? Why weren't papers chosen from the Scopus citation database, which covers Wiley, Spring-Linker, and Elsevier Science (Science direct) journal publishers?

(6) Line 105: The term "partial unfolding of" is not bracketed.

(7) Line 105-107: The solubility is typically increased only by very brief exposures to acoustic cavitation with ultrasound probes, though it can be increased by applying ultrasound treatment in water baths. Let the authors pay attention to this statement and modify it as a possibility, not as something basic and fundamental.

(8) Pay attention and describe how surface solubility and hydrophobicity (H0) are increased by high-intensity ultrasound. Would the authors draw a correlation between these two properties of proteins that are remarkably comparable to one another?

(9) Line 214: The term "H0" needs to be written like an italic font of "H" and a regular font of "zero" in subscript. 

(10) Tables 11, 12 : The font size of the insert text is larger than the above tables (from 1 to 10). Make the all tables uniform.

(11) Line 368: The term "enzymolysis" change with another more apropriated term "enzymatic hydrlysis".

(12) Weather is conceivable for the authors to rephrase their conclusion section by drawing a general (concluded) influence on each examined propeties (techno-functional, structural, microstructural, enzymatic hydrolysis) of cereal protein?

(13) It is advised that the authors recheck the main text during the revision to make this manuscript more readable.

Author Response

Response to reviewer #3's comments

Totally speaking, this study is well designed. Firstly, this paper briefly discussed the ultrasonication effects on the enhancement of physicochemical, microstructure, and techno-functional characteristics of the protein, particularly focusing on the cereal proteins such as corn, barley, and sorghum. Elsevier Science, Web of Science, Wiley, and Spring-Linker databases were referred to search all connected articles from 2000 to 2022. The presented and discussed results have indicated that ultrasonic parameters and the type of sonication could affect (increase or decrease) the functional, structural, enzymatic, and digestive properties of cereal proteins. This study was suitable for this journal. However, there are two issues needed to be clarified.

Q1: Lines 6 and 7: Please include the matching e-mail addresses of the co-authors listed above, as well as their initials in parenthesis.

Answer: Thanks for your review and kind advice. We have revised the contents. We have added the matching e-mail addresses of the co-authors listed above and their initials in parenthesis. The contents were showed by blue font. (Lines ,7,8,9,11)

Q2: Specify the results acquired from this study of literature data in the work's Abstract. As a nutshell, the entire abstract is theoretical. For example, to identify a set of variables that lead to enhanced functional properties and conformational changes in one kind of protein molecule. Also, which type of sonication is best for causing beneficial modifications in cereal proteins?

Answer: Thanks for your review and kind advice. According to the literature data, high-intensity ultrasound is the best for causing beneficial modifications in cereal proteins. We reviesd the Abstract of our work. The contents were showed by blue font. “High-intensity ultrasound can break the hydrogen bond, electrostatic interaction and hydration between cereal protein molecules, expose the hydrophobic groups buried in the molecules, and change the spatial structure of proteins. First, it increases the enzyme binding sites and improve the speed of enzymatic reaction. Secondly, the change of spatial structure exposes more amino acid functional groups and changes the function of cereal protein. thirdly, the increase of surface hydrophobicity and the exposure of hydrophobic groups make the protein structure loose and make cereal protein more easily hydrolyzed.” (Abstract)

Q3: Line 53: I suggest that the authors use the term "high-intensity ultrasound" or "sonication technology" because it has been proven that only high-intensity, low-frequency ultrasound can lead to structural changes in proteins at the quaternary and tertiary structure levels.

Answer: Thanks for your review and kind advice. We have use the term "high-intensity ultrasound" or "sonication technology" in the paper.

Q4: High-intensity ultrasound must be explained in more detail in the Introduction section, along with what kind of technology it is (frequency range, power range, and amplitude range) and the devices that are utilized to achieve it. It would also be beneficial to discuss some of the drawbacks of other non-thermal technologies (such as microwaves or high pressure) as well as the benefits of ultrasonic technology.

Answer: Thanks for your review and kind advice. We have added much more detail about high-intensity ultrasound in the Introduction section. “The application of ultrasound technology in the food industry has attracted much attention. Ultrasound is a mechanical wave with a frequency higher than the que of human hearing (>16 kHz) and can be classified into two types, low-intensity high-frequency and high-intensity low-frequency ultrasound. Low-intensity high-frequency ultrasound (100 kHz-1 MHz, intensity <1 W/cm2) is most often used as a non-destructive analytical testing technique to provide information on the physicochemical properties of food, such as hardness, ripeness, brix and acidity. While high intensity low frequency (100 kHz -1 MHz, intensity <1W/cm2) ultrasound can be used to physically or chemically alter the properties of food. Compared with non-thermal treatment such as microwave and ultra-high pressure, according to the properties of ultrasound, at room temperature and pressure ultrasound can rapidly break the plant cell wall, so that plant proteins can enter the extraction medium such as aqueous solution quickly and efficiently, and accelerate the physical or chemical modification of proteins, thus improving protein solubility and promoting protein conformation unfolding.” The contents were showed by blue font. (Introduction Section)

Q5: Lines 90-91: Why did the authors decide to use only the 181-paper Web of Science indexed citation database? Why weren't papers chosen from the Scopus citation database, which covers Wiley, Spring-Linker, and Elsevier Science (Science direct) journal publishers?

Answer: Thanks for your review and kind advice and pointing out the question. Because this review was mainly focused on articles on the Web of Science, so we decided to use only the 181-paper Web of Science. We will choose from the Scopus citation database when we write the following review.

Q6: Line 105: The term "partial unfolding of" is not bracketed.

Answer: Thanks for your review and kind advice and pointing out the mistakes. We have put "partial unfolding of" in bracket in Line 133-134. (Lines 133,134)

Q7: Line 105-107: The solubility is typically increased only by very brief exposures to acoustic cavitation with ultrasound probes, though it can be increased by applying ultrasound treatment in water baths. Let the authors pay attention to this statement and modify it as a possibility, not as something basic and fundamental.

Answer: Thanks for your review and kind advice and pointing out the mistakes. We have paid attention to the statement and revised in the paper. I have revised the presentation. (Lines 133)

Q8: Pay attention and describe how surface solubility and hydrophobicity (H0) are increased by high-intensity ultrasound. Would the authors draw a correlation between these two properties of proteins that are remarkably comparable to one another?

Answer: Thank you for your question. We have added the content in the paper. (With the prolongation of ultrasonic treatment, the spatial structure of the protein changes, exposing more hydrophobic amino acid residues and increasing the hydrophobicity of the protein surface. This is because ultrasonic waves can break the hydrogen bonds, electrostatic interactions and hydration between protein molecules, exposing the hydrophobic groups buried inside the molecules.)  (Lines 272-277)

Q9: Line 214: The term "H0" needs to be written like an italic font of "H" and a regular font of "zero" in subscript. 

Answer: Thanks for your review and kind advice and pointing out the mistakes. We have revised the correct format of this journal, which were showed by blue font. (Lines 265, 267, 269, 282,283. Table 4)

Q10: Tables 11, 12 : The font size of the insert text is larger than the above tables (from 1 to 10). Make the all tables uniform.

Answer: Thanks for your review and kind advice. We have revised the font size of the insert text and made the all tables uniform. (Tables 11, 12)

Q11: Line 368: The term "enzymolysis" change with another more apropriated term "enzymatic hydrlysis".

Answer: Thanks for your review and kind advice and pointing out the mistakes. We have changed " enzymolysis " into “enzymatic hydrlysis" in Line 449.

Q12: Whether is conceivable for the authors to rephrase their conclusion section by drawing a general (concluded) influence on each examined propeties (techno-functional, structural, microstructural, enzymatic hydrolysis) of cereal protein?

Answer: Thanks for your review and kind advice. We have revised the content in the paper, and added the conclusion section that influenced on each examined propeties of cereal protein. We have already marked it out by red font. (Section 3.1 to Section 3.10)

Q13: It is advised that the authors recheck the main text during the revision to make this manuscript more readable.

Answer: Thanks for your review and kind advice. We have recheck the main text and rewritten some sentences in the paper. We have already marked it out by purple font.
